# Interspecies nutrient extraction and toxin delivery between bacteria

Ofer Stempler[1], Amit K. Baidya[1], Saurabh Bhattacharya[1], Ganesh Babu Malli Mohan[1,2], Elhanan Tzipilevich[1], Lior Sinai[1], Gideon Mamou[1] & Sigal Ben-Yehuda[1]

Bacteria have developed various mechanisms by which they sense, interact, and kill other bacteria, in an attempt to outcompete one another and survive. Here we show that *Bacillus subtilis* can kill and prey on *Bacillus megaterium*. We find that *Bacillus subtilis* rapidly inhibits *Bacillus megaterium* growth by delivering the tRNase toxin WapA. Furthermore, utilizing the methionine analogue L-azidohomoalanine as a nutrient reporter, we provide evidence of nutrient extraction from *Bacillus megaterium* by *Bacillus subtilis*. Toxin delivery and nutrient extraction occur in a contact-dependent manner, and both activities are abolished in the absence of the phosphodiesterase YmdB, shown previously to mediate intercellular nanotube formation. Furthermore, we detect the localization of WapA molecules to nanotubes. Thus, we propose that *Bacillus subtilis* utilizes the same nanotube apparatus in a bidirectional manner, delivering toxin and acquiring beneficial cargo, thereby maximally exploiting potential niche resources.

[1] Department of Microbiology and Molecular Genetics, Institute for Medical Research Israel-Canada, The Hebrew University-Hadassah Medical School, POB 12272, The Hebrew University of Jerusalem, 91120 Jerusalem, Israel. [2] Present address: California Institute of Technology, Jet Propulsion Laboratory Biotechnology and Planetary Protection Group, M/S 89-102, 4800 Oak Groove Dr., Pasadena, California 91109, USA. Ofer Stempler, Amit K. Baidya and Saurabh Bhattacharya contributed equally to this work. Correspondence and requests for materials should be addressed to S.B.-Y. (email: sigalb@ekmd.huji.ac.il)

In natural microbial communities, each bacterium possesses an array of mechanisms designed to constantly interact with its microbial neighbors. While some mechanisms can act from afar, enabling the delivery of antimicrobial compounds into the shared milieu (e.g., quorum sensing, antibiotic production, membrane vesicle release), others require a tight physical connection between the interacting cells (e.g; refs. [1–3]). A well studied example for contact mediated interaction is the type VI secretion system, which plays a key role both in bacterial competition and in bacterial infection. This system assembles as a tubular puncturing device, through which bacteria can deliver toxic molecules directly into prokaryotic or eukaryotic target cells[4]. Another strategy of intercellular interaction is exemplified by the contact-dependent inhibition phenomenon. In this mode, cells inhibit growth of their neighbors using a system, which comprises an outer-membrane toxin protein, its corresponding transporter and a cognate immunity protein[5, 6]. The diversity and complexity of direct intercellular interaction systems emphasizes the multiple strategies employed by bacteria to interact with their surroundings.

Some contact-dependent mechanisms are designed not necessarily to actively inhibit rivals, but rather to insure the bacterium's own survival by exchanging vital compounds between the cells (e.g; refs. [7, 8]). We have previously shown that such communication can be mediated by the production of small 'tube-like' membranous structures, formed among adjacent bacterial cells, which we previously termed nanotubes[9]. Utilizing the Gram-positive bacterium *Bacillus subtilis* (*Bs*) as a model organism, we demonstrated that nanotubes can allow for intercellular transfer of cytoplasmatic molecules from one bacterium to another, resulting in transient acquisition of nonhereditary antibiotic resistance. Cryo-EM analysis further revealed that nanotubes are composed of chains of membranous segments harboring a continuous lumen[10]. Similar membranous structures were implicated in long range extracellular electron transport in *Shewanella oneidensis* MR-1[11]. In addition, two recent studies have shown that the bacterial pairs *Acinetobacter baylyi* and *Escherichia coli*, as well as *Clostridium acetobutylicum* and *Desulfovibrio vulgaris*, can cross feed each other in the absence of vital nutrients, utilizing nanotube-like structures[12, 13].

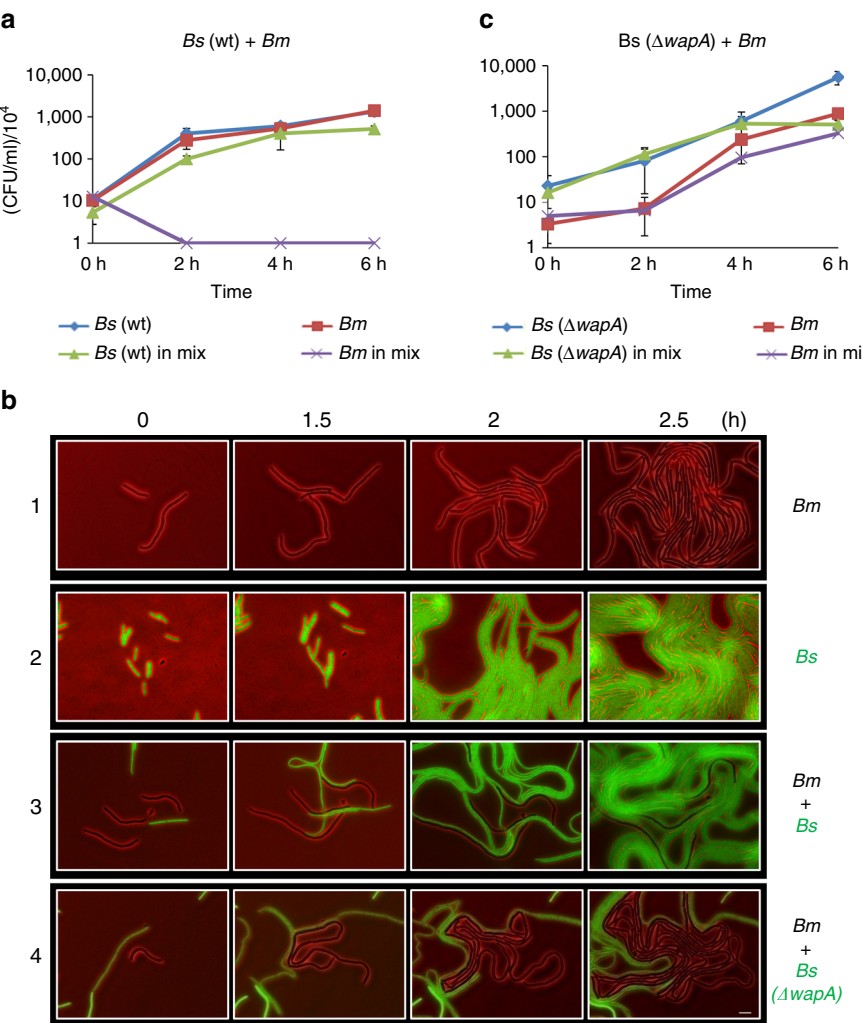

**Fig. 1** *Bm* inhibition by *Bs* is mediated by the WapA toxin. **a** *Bs* (PY79, wt) and *Bm* (OS2) were grown separately or in a mixture in LB medium at 37 °C. Cells were plated for CFUs at the indicated time points. Each point represents an average value and s.d. of three independent experiments. **b** Representative time lapse microscopy images of (1) *Bm* (OS2), (2) *Bs* (AR16: amyE::P_rrnE-gfp), (3) a mixture of *Bm* (OS2) and *Bs* (AR16), and (4) a mixture of *Bm* (OS2) and *Bs* (OS24: ΔwapA, sacA::P_veg-mCherry). Shown are overlay fluorescence from GFP or mCherry (*green*) and phase contrast (*red*) images, captured at the indicated time points. *Bm* cells are shown in *black* while *Bs* cells are shown in *green*. *Scale bar* represents 5 μm. **c** *Bs* (OS16::ΔwapA) and *Bm* (OS2) were grown separately or in a mixture and processed as in **a**. Each point represents an average value and s.d. of three independent experiments

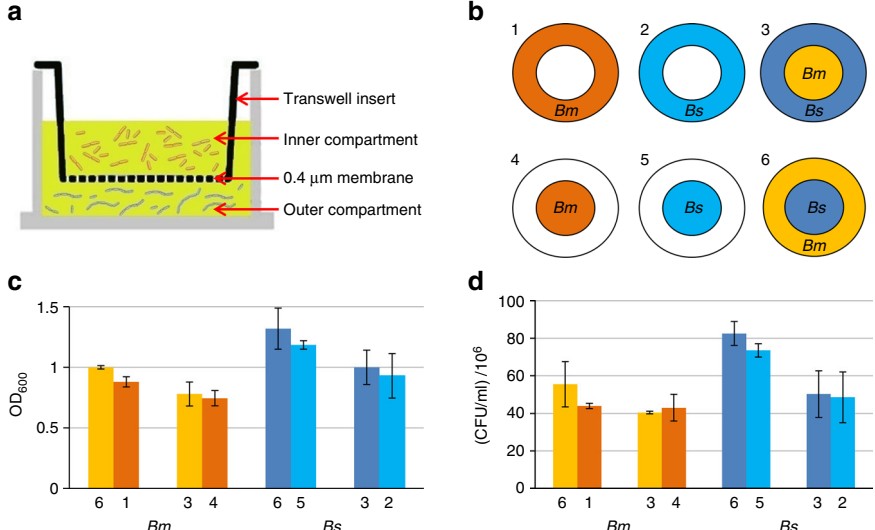

**Fig. 2** *Bs* inhibits *Bm* growth in a contact-dependent manner. **a** Illustration of a single Transwell chamber. The medium is shared between the inner and outer compartments, while the cells cannot pass the 0.4 μm membrane barrier. **b** *Bs* (PY79) and *Bm* (OS2) cells were grown in Transwell chambers containing LB as indicated by the layout. Compartments containing *Bs* are shown in *blue hues*, while compartments containing *Bm* are shown in *yellow hues*. *White circles* represent compartments containing LB only. **c, d** Shown are OD$_{600}$ (**c**) and CFUs (**d**) of cells grown in Transwell chambers for 6 h. The numbers and colors correspond to the layout described in **b**. Each column represents an average value and s.d. of three independent experiments

Here we show that *Bs* delivers the WapA toxin to, and extracts nutrient from *Bacillus megaterium* (*Bm*), which we interpret as two parallel strategies to acquire territory and nutrients. We further present evidence that these predatory activities are contact-dependent and require the phosphodiesterase YmdB, previously shown to be required for nanotube formation[10].

## Results

**Bs inhibits Bm growth in a contact-dependent manner.** To study how different species impact each other, we examined the interaction between the soil bacterium *Bs* PY79[14] and natural soil isolate bacteria. We enriched for spore forming bacteria derived from a soil niche, and the isolated strains were identified using 16S rDNA analysis. Among the isolated strains was *Bm*, which henceforth was utilized for interspecies analysis. We first carried out a competition assay, in which *Bs* was co-cultured with the wild isolate *Bm* in liquid medium (1:1 ratio), and growth was followed by colony forming units (CFUs). Due to their distinct morphological differences, the colonies of the two *Bacillus* species could be easily distinguished (Supplementary Fig. 1A). We found that *Bm* failed to form colonies after merely 2 h of co-incubation (Fig. 1a; Supplementary Fig. 1B), indicating that *Bs* greatly inhibits its growth, although when grown separately the two *Bacilli* exhibit a very similar growth rate (Supplementary Fig. 1C). To further analyze the antagonistic effect produced by *Bs*, we examined the consequence of growing *Bm* and GFP labeled *Bs* on a solid surface, in a confined space, using time lapse fluorescence microscopy. We observed that when *Bs* encounters *Bm*, an abrupt halt of *Bm* growth and division occurs, culminating with *Bs* occupying the majority of the field (Fig. 1b, 1–3), consistent with the CFU viability assay (Fig. 1a). To further inspect the observed *Bm* growth halt, we supplemented the medium with propidium iodide (PI), which preferentially penetrates and stains cells that lost their membrane integrity. The analysis revealed that the inhibited *Bm* cells remained unstained, indicating that they did not lyse and maintained an intact membrane (Supplementary Fig. 2). As a comparison, damaging specific cells using highly accurate laser ablation was followed by PI accumulation in the irradiated cells (Supplementary Fig. 2)[15].

To examine the nature of the observed inhibition, we aimed to determine if it is mediated by an antagonistic factor secreted into the medium, or requires direct cell to cell contact. To address this question, the two *Bacilli* were placed in a Transwell plate, consisting of double chambered wells, separated by a membrane. The membrane allows the passage of small molecules, while the cells remain compartmentalized (Figs. 2a, b). Following OD$_{600}$ and CFU values for *Bm* and *Bs*, located at the different compartments, revealed that *Bm* growth was not affected by the presence of *Bs* in the adjacent chamber, even after 6 h of co-growth (Figs. 2c, d). However, when *Bm* and *Bs* were co-incubated in the same compartment *Bm* growth inhibition was restored (Supplementary Fig. 3A). These results indicate that inhibition of *Bm* by *Bs* is not due to an antagonistic molecule secreted into the shared medium, but rather demands physical contact for its occurrence.

**Bs toxicity is mediated by delivery of WapA toxin to Bm.** We next searched for potential toxins encoded by *Bs* that could cause *Bm* growth inhibition. It has been previously reported that *Bs* encodes a *wapA-wapI* toxin-antitoxin module, in which the WapA toxin was found to cleave tRNases, perturbing translation. WapA was shown to act in an intercellular contact-dependent manner, causing toxicity to neighboring *Bs* cells lacking the protective antitoxin[16]. We therefore reasoned that WapA could also act in an interspecies manner, accounting for the *Bs* toxic activity against *Bm*. To test this idea, we conducted a liquid competition assay between *Bs*, lacking *wapA*, and *Bm*. Indeed, *Bs* mutant cells failed to inhibit *Bm* growth as indicated by the increase of *Bm* CFU in the mixed culture (Fig. 1c). Consistently, following the growth of *Bm* and *Bs* (Δ*wapA*) by time lapse microscopy revealed that *Bm* was indeed able to grow in the presence of the mutant *Bs* (Fig. 1b). Furthermore, introducing the *Bs* wapI antitoxin gene into *Bm* conferred full immunity to *Bm* in mixed cultures (Supplementary Fig. 3B, C). Taken together, our results are consistent with the view that *Bs* inhibits growth of the wild *Bm* isolate in a contact-dependent manner by delivering the WapA toxin.

**Bm inhibition by Bs is YmdB-mediated.** We next searched for additional potential genes encoded by *Bs* that could be involved in

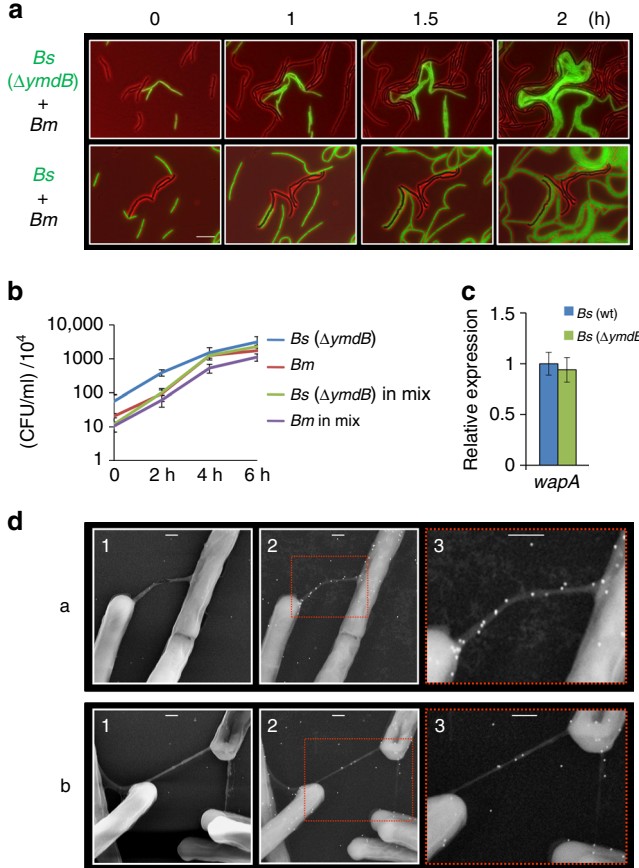

**Fig. 3** *ymdB* mediates *Bm* inhibition by *Bs*. **a** Representative time-lapse microscopy images displaying a mixture of *Bs* (OS23: Δ*ymdB, sacA*::P$_{veg}$-*mCherry*) and *Bm* (OS2) (*upper panels*), and a mixture of *Bs* (AR16: *amyE*::P$_{rrnE}$-*gfp*) and *Bm* (OS2) (*lower panels*). Shown are overlay fluorescence from mCherry or GFP (*green*) and phase contrast (*red*) images, captured at the indicated time points. *Bm* cells are shown in *black* while *Bs* cells are shown in *green*. *Scale bar* represents 10 μm. **b** *Bs* (GB61: Δ*ymdB*) and *Bm* (OS2) were grown separately or in a mixture in LB medium at 37 °C. Cells were plated for CFUs at the indicated time points. *Bs* and *Bm* were differentiated by colony morphology. Each point represents an average value and s.d. of three independent experiments. **c** RNA was isolated from *Bs* (PY79) and *Bs* (GB61: Δ*ymdB*) cells grown to the mid logarithmic phase and expression of *wapA* was determined by qRT-PCR. Transcript levels are relative to *Bs* (PY79). Each bar represents an average value and s.d. of three independent experiments. **d** *Bs* (SH29): *wapA*-2×HA, Δ*ymdB*, *amyE*::P$_{hyper-spank}$-*ymdB*, Δ*hag*) cells harboring *wapA*-2xHA were spotted onto EM grids. Cells were subjected to immuno-gold HR-SEM using primary antibodies against HA and secondary gold-conjugated antibodies. Samples were not coated before observation. Shown are two examples of WapA-2xHA localization (*white dots*) to nanotubes: intimate intercellular nanotube (a) and a nanotube network view b. HR-SEM images were acquired using TLD-SE (Through lens detector-secondary electron) (1) and vCD (low-kV high-contrast detector) (2). An enlargement of the *boxed* regions in (2) is shown in (3). *Scale bars* represent 200 nm

the *Bs-Bm* interspecies interaction. We reasoned that mutants in such gene(s) would fail to inhibit *Bm* growth, similarly to the *Bs* mutant lacking *wapA*. We conducted a screen using different *Bs* mutants, lacking a potential gene or genes that might prove accountable for the inhibition. Our survey included 27 mutants (Supplementary Table 1) deficient mainly in biofilm formation (e.g, *tasA*, *sinI*, *sinR*, *ymdB*), chemotaxis (*sigD*), membrane metabolism (e.g, *ywnE*, *yerQ*, *psd*), as well as a mutant lacking the

major developmental regulator Spo0A. Each of the investigated mutant strains was mixed with *Bm*, and growth was inspected using time lapse microscopy. Our screen revealed that only the *ymdB* mutant failed to inhibit *Bm* growth (Fig. 3a), yielding results similar to that of the *wapA* mutant (Fig. 1b). Consistent with the microscopy screen, *ymdB* mutant failed to inhibit *Bm* growth when grown similarly in liquid LB medium (Fig. 3b). YmdB, a conserved calcineurin-like phosphodiesterase, was previously shown using microarray analysis to affect the mRNA levels of multiple genes; however, no modifcation in *wapA* transcript level was detected[17, 18]. We re-examined this possibility utilizing quantitative reverse transcription PCR (qRT-PCR) and fluorescence from P$_{wapA}$-GFP reporter. Both methods confirmed that *ymdB* does not influence *wapA* expression levels (Fig. 3c; Supplementary Fig. 4A). YmdB has been previously implicated in biofilm formation[18], and was shown by our laboratory to be required for cAMP production, proper colony development, and the formation of intercellular nanotubes[10, 15]. Furthermore, WapA protein was previously detected in the nanotube enriched biochemical fraction[10]. To investigate the possibility that WapA is directly associated with nanotubes, *Bs* cells harboring WapA tagged with HA (human influenza hemagglutinin) epitope, were subjected to non-coated immuno- High Resolution Scanning Electron Microscopy (HR-SEM) analysis utilizing anti-HA antibodies and gold-labeled secondary antibodies. Indeed, WapA molecules were detected along intercellular nanotubes branching from cells and were also scattered over the cell surface (Fig. 3d). The immuno-gold signal was not detectable from cells or nanotubes of *Bs* lacking HA-tagged WapA (Supplementary Fig. 4B). Thus, our data support the view that *ymdB* is required for inhibiting the growth of the opponent *Bm* by delivering the WapA toxin via nanotubes.

**Methionine extraction from *Bm* by *Bs*.** So far we revealed that *Bs* is capable of delivering the WapA toxin into *Bm*, killing a potential competitor within the niche, and that this antagonistic activity also requires the phosphodiesterase YmdB. We next examined whether *Bs* can also act inversely, extracting nutrients from *Bm* for its own benefit. Evidence for mutual cross feeding between different auxotrophic strains has been previously reported (e.g., ref. [12]). To observe such potential molecular delivery, we devised a strategy utilizing the methionine (Met) analogue, L-azidohomoalanine (AHA), as a nutrient reporter. AHA is incorporated into newly synthesized proteins, and can be readily monitored using fluorescence microscopy, by tagging the AHA azide group with a fluorophore using copper catalyzed click reaction[19]. *Bs* or *Bm* cells were pre-grown in the presence of Met in order to repress the expression of Met biosynthesis and boost the expression of Met transporters[20]. Cells were then washed and incubated for 1.5–2 h with AHA as a substitute for Met. The subsequent click reaction revealed that cells of both *Bacilli* were heavily labeled by AHA, as indicated by the strong fluorescence signal emitted from the cells (Fig. 4a). However, when the cells of both strains were pre-grown in the absence of Met, and then similarly washed and incubated with AHA, no fluorescence signal was detected (Fig. 4b). These results indicate that when Met is scarce, both strains keep producing endogenous Met for their own growth, which is preferable over AHA[21], and therefore remain unstained (Fig. 4d). Consistently, utilizing a Met auxotrophic *Bs* strain (Δ*metE*) resulted in an increase in AHA uptake, as indicated by a stronger fluorescence signal compared to the prototroph wild type strain (Fig. 4c; Supplementary Fig. 5).

To detect molecular exchange between the two *Bacilli*, *Bm* pre-grown with or without Met was mixed with *Bs* (Δ*metE*), and the mixture was supplemented with AHA instead of Met and

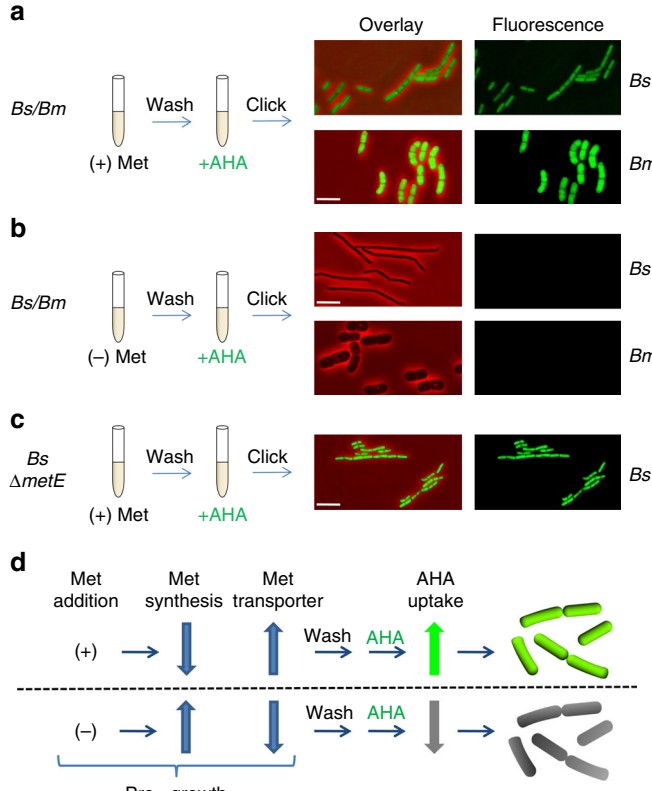

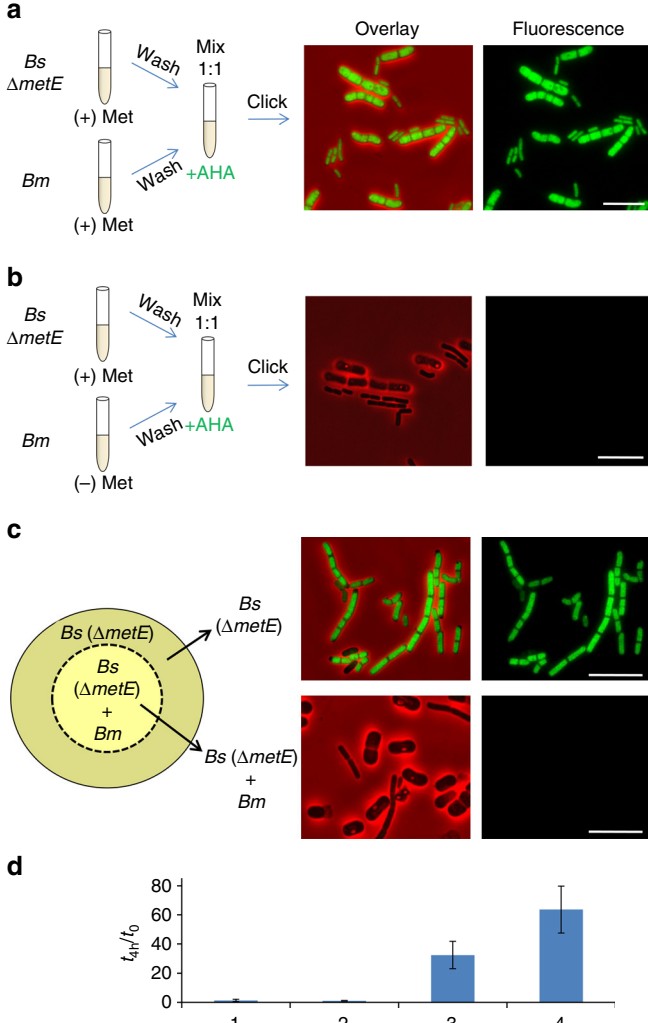

**Fig. 4** Pre-incubation with Met affects AHA uptake. **a**, **b** *Bs* (PY79) or *Bm* (OS2) were grown over night in S7 medium in the presence (**a**) or the absence (**b**) of Met (50 μg/ml). Cells were then washed in PBS×1 and resuspended in S7 supplemented with AHA (1 mM). Cells were incubated for additional 2 h, underwent click reaction and visualized by fluorescence microscopy. Shown are overlay images of fluorescence from AHA (*green*) and phase contrast (*red; left panels*), and AHA fluorescence alone (*right panels*). **c** Met auxotroph *Bs* (LS5: *ΔmetE*) was grown over night with Met and processed as described in **a**. **d** A schematic model of exogenous Met effect on *Bacilli* Met transporters and endogenous Met synthesis. Pre growth in the presence of external Met results in AHA uptake, whereas pre-growth in the absence of external Met restricts AHA uptake. *Fluorescence images* were normalized to the same intensity range for each strain. *Scale bars* represent 3 μm

incubated for 1.5–2 h. As expected, when *Bs* (*ΔmetE*) was mixed with *Bm*, pre-grown with Met, the two *Bacilli* were labeled with AHA (Fig. 5a). In contrast, when *Bs* (*ΔmetE*) was mixed with *Bm*, pre-grown in the absence of Met, both *Bacilli* cells remained unlabeled (Fig. 5b), indicating the lack of AHA uptake. Since *Bs* (*ΔmetE*) must obtain external Met/AHA for its survival, the lack of AHA labeling suggests that *Bs* (*ΔmetE*) was able to acquire Met from *Bm*. Conceivably; the expression of the Met transporters in the *Bs* (*ΔmetE*) was decreased during the co-incubation period, resulting in unlabeled cells. Consistent with this idea, *Bs* (*ΔmetE*) could grow and divide in a medium lacking Met, only when co-cultured with *Bm* (Fig. 5d). We interpret these results to conclude that *Bs* (*ΔmetE*) acquires Met from its neighboring *Bm* either from the shared medium or in a contact-dependent manner. To test these possibilities, we designed a Transwell experiment in which *Bs* (*ΔmetE*) was grown simultaneously in two separate compartments in medium containing AHA. One compartment contains *Bs* alone, while the adjacent compartment contains a mixture of *Bs* and *Bm*, the latter pre-grown in the absence of Met. In such system, the two cultures share the same AHA containing medium (Fig. 5c). Following

**Fig. 5** *Bs* extracts Met from its neighboring *Bm*. **a**, **b** *Bm* (OS2) was pre-grown in the presence (**a**) or the absence (**b**) of Met (50 μg/ml), washed in PBSx1, mixed with *Bs* (LS5: *ΔmetE*) and resuspended in S7 supplemented with AHA (1 mM). Cells were incubated for additional 1.5 h, underwent click reaction and visualized by fluorescence microscopy. Shown are overlay images of fluorescence from AHA (*green*) and phase contrast (*red; left panels*), and AHA fluorescence alone (*right panels*). *Fluorescence images* were normalized to the same intensity range. **c** *Bm* (OS2), pre grown in the absence of Met, and *Bs* (LS5: *ΔmetE*) were mixed and incubated in the inner Transwell compartment. For a comparison, *Bs* (LS5: *ΔmetE*) was incubated in the outer Transwell compartment. Both the mixture in the inner well, and the *Bs* grown alone in the outer well, shared the same S7 medium containing AHA (1 mM). After 2 h of incubation, the cells underwent click reaction and visualized by fluorescence microscopy. Shown are overlay images of fluorescence from AHA (*green*) and phase contrast (*red; left panels*), and AHA fluorescence alone (*right panels*). Fluorescence images were normalized to the same intensity range. **d** Shown are CFU ratios at $t_{4-h}/t_0$ of the following cultures: (1) *Bs* (LS5: *ΔmetE*) grown in the absence of Met. (2) *Bs* (OS21: *ΔmetE*, *ΔymdB*) co-cultured with *Bm* (OS2) in the absence of Met. (3) *Bs* (LS5: *ΔmetE*) co-cultured with *Bm* (OS2) in the absence of Met. (4) *Bs* (LS5: *ΔmetE*) grown in the presence of Met. *Bm* OS2 cells were pre-grown in the absence of Met. Shown are average values and s.d. of three independent experiments. *Scale bars* represent 10 μm

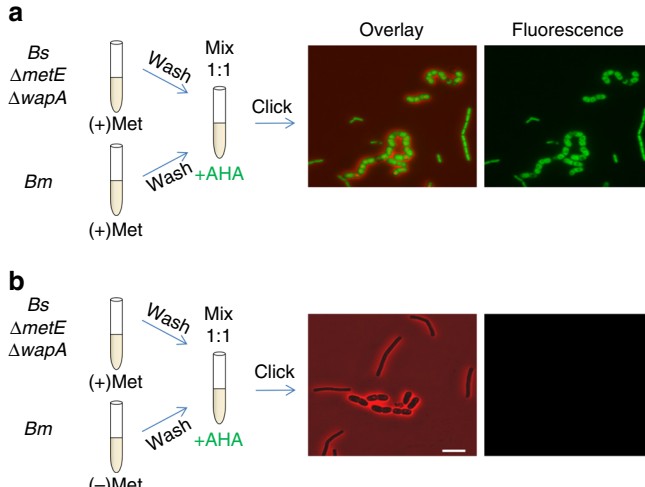

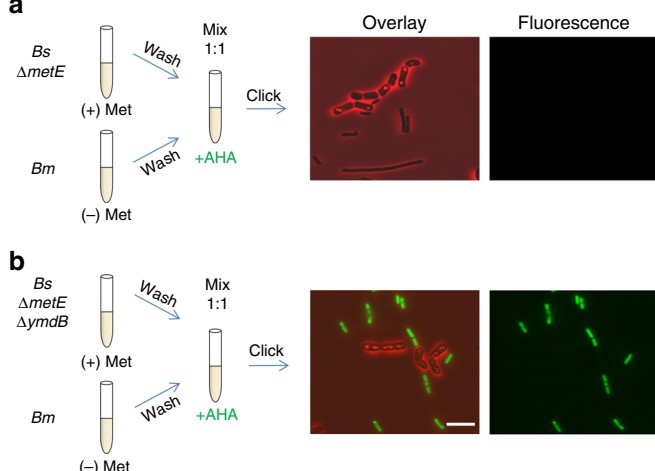

**Fig. 6** WapA delivery is not a prerequisite for Met extraction. **a**, **b** *Bm* (OS2) was grown over night in S7 medium in the presence (**a**) or the absence (**b**) of Met (50 µg/ml). Cells were then washed in PBSx1, mixed with *Bs* (OS25: Δ*metE*, Δ*wapA*) and resuspended in S7 supplemented with AHA (1 mM). Cells were incubated for additional 1.5 h, underwent click reaction and visualized by fluorescence microscopy. Shown are overlay images of fluorescence from AHA (*green*) and phase contrast (*red*; *left panels*), and AHA fluorescence alone (*right panels*). *Scale bar* represents 5 µm

growth, cells from each chamber underwent click reaction separately. Our analysis revealed that *Bs* (Δ*metE*), grown alone, exhibited an apparent fluorescence signal from AHA, indicating that it could not acquire Met from *Bm* via the shared medium. On the other hand, neither of the strains in the mixture, located in the adjacent compartment, displayed any detectable fluorescence signal (Fig. 5c). These results show that *Bs* is capable of extracting Met from *Bm*, and that a direct contact must exist to facilitate this molecular extraction.

We next sought to explore whether the killing of *Bm* is a prerequisite for nutrient extraction by *Bs*. We therefore, co-cultured *Bm*, pre-grown in the presence or the absence of Met, with *Bs* (Δ*metE*) lacking the *wapA* gene. The absence of WapA did not affect the ability of *Bs* to acquire Met from *Bm*, as the results were similar to those of the *Bs* (Δ*metE*) parental strain harboring intact WapA (Fig. 6). Thus, although Met extraction and toxin delivery are both contact-dependent phenomena, they operate autonomously.

**YmdB is required for Met extraction from *Bm* by *Bs*.** A plausible mechanism that could account for the contact mediated exchange of nutrients between the two *Bacilli* might involve the YmdB phosphodiesterase, and particularly its role in nanotube formation. We therefore examined whether the ability of *Bs* to extract nutrients from *Bm* is also YmdB mediated. To test this possibility, *Bs* (Δ*metE*) harboring Δ*ymdB* was incubated for 1.5 h with *Bm* (pre-grown in the absence of Met) in the presence of AHA, and the mixture was subjected to click reaction. The results obtained were explicit, *Bs* (Δ*metE*) cells lacking *ymdB* were heavily labeled with AHA (Fig. 7b), while *Bs* (Δ*metE*) cells harboring intact *ymdB* remained unlabeled (Fig. 7a), similarly to the results displayed in Fig. 5b. Consistent with these results, *Bs* (Δ*metE*) cells lacking *ymdB* were unable to grow and divide in a medium lacking Met, when co-cultured with *Bm* (Fig. 5d). These findings show that *Bs* lacking *ymdB* is deficient in extracting the favorable Met from *Bm*, and consequently acquires AHA from the medium.

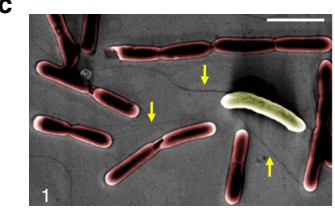
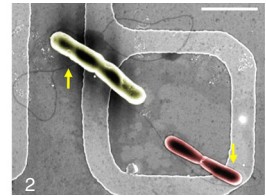

**Fig. 7** *ymdB* is required for Met extraction. **a**, **b** *Bm* (OS2) was pre grown in the absence of Met (50 µg/ml), washed in PBS×1, mixed with *Bs* (LS5: Δ*metE*) (**a**) or *Bs* (OS21: Δ*metE*, Δ*ymdB*) (**b**), and resuspended in S7 supplemented with AHA (1 mM). Cells were incubated for additional 1.5 h, underwent click reaction and visualized by fluorescence microscopy. Shown are overlay images of fluorescence from AHA (*green*) and phase-contrast (*red*; *left panels*), and AHA fluorescence alone (*right panels*). Fluorescence images were normalized to the same intensity range. *Scale bar* represents 10 µm. **c** *Bs* (GB168: Δ*ymdB*, Δ*hag*, *amyE*::P_{hyper spank}-*ymdB*) and *Bm* (OS2) cells were incubated at low density in LB medium for 1 h, and then fixed and visualized by HR-SEM without coating. Shown are HR-SEM images (×15,000) of (1) a single *Bm* cell (*yellow*) linked via nanotubes to several *Bs* cells (*red*), and (2) a single nanotube linking neighboring *Bs* and *Bm*. *Arrows* point to nanotube structures. *Scale bars* represent 3 µm

The observation that toxin delivery and nutrient extraction are dependent on YmdB, the findings that both activities are contact-dependent, and the localization of WapA to nanotubes, suggest that interspecies nanotubes are involved in these processes. To examine the existence of nanotube connections between the two investigated strains, we utilized a method enabling the visualization of nanotubes in living cells at low cell density using a fluorescent membrane dye[10]. Nanotubes could be observed in such conditions upon intense fluorescent exposures. By exploiting this approach, we could detect long nanotubes emerging from *Bs*, connecting neighboring cells (Supplementary Fig. 6A). Elongated nanotubes were also observed to link two *Bm* cells (Supplementary Fig. 6A), indicating that *Bm* is also capable of forming tubes. Furthermore, when the two *Bacilli* were mixed, nanotubes bridging *Bs* and *Bm* cells were visible (Supplementary Fig. 6A). To better detect these interspecies connections, the mixture was visualized using non-coated HR-SEM[10]. We have previously shown that nanotubes observed by fluorescence microscopy correlated with nanotube structures observed by non-coated HR-SEM[10]. Indeed, a network of long distance and short intimate nanotubes was evidently observed to connect the two *Bacilli* (Fig. 7c; Supplementary Fig. 6B, C). These results reinforce the view that nanotubes could serve as the route for

molecular exchange between the two species. We suggest that the lack of YmdB perturbs nanotube formation, which in turn impairs *Bs* ability to deliver WapA to *Bm* and extract its Met.

## Discussion

Here we show that *Bs* employs two complementing approaches in an attempt to antagonize its *Bm* opponent, both by delivering the WapA toxin to halt its growth and by simultaneously preying on its vital nutrient components. We demonstrate that both activities are mediated by YmdB, but operate independently. Since YmdB is required for nanotube formation[10], WapA was found to be localized to nanotube structures, and nanotubes were found to bridge *Bs* and *Bm*, we propose that nanotubes facilitate this bidirectional molecular trafficking, and that *Bs* could both deliver and acquire molecules via the same path.

Nutrient exchange leading to cross feeding among bacterial cells has been known for many years, and was utilized as a basic tool for deciphering bacterial metabolic pathways (e.g, refs. [22–24]). However, only in recent studies the requirement of cell to cell contact for such exchange was substantiated, as in some cases it was found to be associated with nanotube like structures bridging distant bacterial species[12, 13]. Interestingly, long extracellular extensions involved in nutrient delivery from afar were exemplified to be employed by the Gram-negative *Delftia* sp. In response to a unique carbon source, this bacterium produces membrane vesicle chains, termed nanopods, extending to long distances[25], suggesting that nanopods are utilized for long distant acquisition of specific nutrients. The ability to directly extract vital molecules from the surrounding bacteria holds an enormous beneficial potential, as the bacterium not only eliminates its rival, but also has the key to extract valuable nutrients from its opponent reservoir in a direct and efficient way. Thus, after inhibiting the growth of its niche rival, the bacterium can further benefit from exploiting its opponent's vital resources. Bacteria harboring tools to employ this strategy could gain access to otherwise restricted nutrient pools, and therefore could dominate a niche, especially under limited nutritional conditions. Such mechanism could be valuable for species competing for the same territory in multicellular biofilm-like structures[26, 27].

Our results show that the delivery of the WapA toxin is responsible for the observed *Bm* growth inhibition by *Bs*. WapA was shown to be constantly exchanged among nearby *Bs* cells, reducing viability in cells lacking the cognate *wapI* immunity gene[16, 28]; however, its exact delivery mechanism was not elucidated. WapA was shown to carry a secretion signal sequence in its N-terminal, and to contain the toxic RNase activity in its C-terminal part[16, 29–31]. Our data show that nanotubes provide a route for WapA molecular delivery. Yet, it is unclear whether WapA molecules localize over the tube surface and/or inside the tube lumen, as the immuno-SEM analysis utilized in this study may perturb nanotube integrity, and the antibody signal could be obtained from the nanotube lumen or surface. It is possible that WapA is interacting with the nanotube membrane, and that this interaction enables WapA translocation. The substantial abundance and variety of toxin-anti toxin modules, with some bacteria carrying multiple systems in their genome[32, 33], brings about the intriguing possibility that nanotubes provide a path for the transport of diverse toxins.

The possibility of bidirectional molecular flow within nanotubes could be explained by a dual function of the same tube or by the existence of two types of operating tubes, each acting in a unidirectional fashion. We have previously shown that proteins and plasmids, which are relatively large molecules, are delivered among *Bs* cells via nanotubes in a donor to recipient directionality[10]. Here we provide evidence that small molecules,

such as Met, can be acquired via nanotubes, raising the possibility that directionality could be determined by the size of the transported cargo. However, other properties such as electrical charge, shape and hydrophobicity could also affect molecular delivery. So far, the known external apparatuses produced by bacteria such as Type III, IV and VI secretion systems, were shown to operate in a unidirectional fashion from donor to recipient, most commonly by delivering toxic molecules into the targeted cells[34]. Although evidence for molecular extraction of valuable nutrients are accumulating[35], an apparatus that facilitates this extraction has not been assigned. We propose that nanotubes, which operate in a contact-dependent manner, can serve as such an apparatus.

Nutrient exchange among different bacterial species was so far monitored by bacterial growth. Here, we devised a method to visualize and quantify nutrient exchange using the AHA surrogate. This method could be similarly utilized to report molecular exchange among any given bacterial species. Furthermore, this strategy can be extended to determine flow directionality of additional labeled molecules, such as nucleotides and various carbon sources. With the growing conceptual view that comprehending interspecies interactions among bacteria is key for better understanding natural bacterial communities, visualizing molecular trade could provide a direct approach to follow molecular paths in complex communities.

## Methods

**Strains and general methods.** *Bs* strains are derivatives of the wild type strain PY79[14], while *Bm* is a wild-type soil isolate (OS2). To isolate *Bm* 1 gm of soil was diluted in LB to $10^{-5}$, incubated 30 min at 80 °C, plated on LB plates, and incubated over night at 30 °C. Colonies were than isolated and sequenced (Hylabs) using 16S rDNA global primers. Bacterial strains and plasmids are listed in Supplementary Table 2, and primers are listed in Supplementary Table 3. All general methods were carried out as described previously[36]. All the experiments were carried out at 37 °C. Transformation into *Bm* OS2 cells was carried out as previously described[37]. In brief, *Bm* OS2 cells were grown up to 1.0 $OD_{600}$ (10 ml). Cells were then washed with electroporation buffer (25% PEG 6000 and 0.1 M sorbitol) and resuspended in 1 ml of electroporation buffer. Electroporation was carried out with 0.1 ml of cells supplemented with 2 µg of plasmid at 1500 V (Bio-Rad). Cells were then diluted into 1 ml of LB and plated on selective antibiotic plate. *wapI* expression was induced by the addition of 5 mg/ml xylose.

**Liquid competition assay.** *Bs* and *Bm* cells were grown separately over night at 23 °C. Cells were then diluted to $OD_{600}$ 0.05 in 8 ml LB, mixed together, and incubated at 37 °C with shaking (200 rpm). CFUs, and $OD_{600}$ were measured as indicated. Strains were differentiated by colony morphology.

**Solid agarose competition assay.** Overnight cultures of *Bs* and *Bm* were diluted to $OD_{600}$ 0.1, mixed together and mounted onto a metal ring (A-7816, Invitrogen) filled with LB agarose (1.5%). Cells were incubated in a temperature controlled chamber at 37 °C, and followed by light microscopy, utilizing Nikon Eclipse Ti microscope. Images were captured using Photometrics CoolSnap HQ[2], and were edited using NIS Elements AR version 4.5.

**Transwell plate assay.** *Bs* and *Bm* cells were grown separately over night at 23 °C. Cells were then diluted to $OD_{600}$ 0.05 and 3 ml of each strain were placed within a Transwell plate (3450 clear, Costar), consisting of 6 wells, each contains two chambers, separated by a 0.4 µm membrane. The Transwell plate was incubated at 37 °C with shaking (30 rpm), and CFUs and $OD_{600}$ were measured as indicated.

**Visualizing AHA uptake.** *Bs* and *Bm* cells were grown over night at 30 °C in S7 minimal medium[38], containing all amino acids or lacking Met. Cells were then diluted to $OD_{600}$ 0.05 and incubated in S7 medium at 37 °C to $OD_{600}$ 0.3. Next, cells were washed in PBSx1 to remove residual Met, diluted to $OD_{600}$ 0.05, and incubated in S7 medium lacking Met, either in monoculture or co-culture, and supplemented with AHA (1 mM). Cells were grown until reaching $OD_{600}$ 0.15 (~ 1.5–2 h), and click reaction was carried out. In brief, cells were centrifuged and washed in PBS×1 to remove residual AHA, resuspended in 50% ethanol for 3 min, followed by a wash in 100% ethanol. Cells were then centrifuged and re-suspended in 230 µl PBSx1 and 12.5 µl of freshly made sodium ascorbate (100 mM; Sigma). Then, mixed separately with 2.5 µl of 50 mM tris(3-hydroxypropyltriazolylmethyl) amine (THPTA, Sigma), 1.25 µl of 20 mM $CuSO_4$ (Sigma), and 3 µl of 2 mM alkyne dye (Alexa Fluor 488 Alkyne, Invitrogen), and left to react for 3 min in

the dark at RT. The mixture was added to the cells, and samples were left in the dark at RT for 30 min. Finally, Cells were centrifuged, washed in PBSx1, and observed by fluorescence microscopy.

**Nanotube detection**. Exponentially growing cells were spotted at low density onto an ITO-coated cover slip[9, 10] and covered with a dialysis membrane. The cover slip was then assembled into a mounting frame (A-7816, Invitrogen) filled with liquid LB supplemented with 1 µg/ml FM 4-64 fluorescent membrane dye (Molecular Probes, Invitrogen). Cells were incubated in a temperature controlled chamber at 37 °C for 1 h. Cells were visualized by Eclipse Ti (Nikon, Japan), equipped with CoolSnap HQII camera (Photometrics, Roper Scientific, USA). System control and image processing were performed with NIS Elements AR 4.3 (Nikon, Japan) and MetaMorph 7.7.5 software (Molecular Devices, USA).

**HR-SEM microscopy for nanotube visualization**. Bs and Bm cells were spotted at low density onto an ITO cover slip, covered with a dialysis membrane, and incubated in LB medium for 1 h. Cells were then left overnight for fixation in 2% glutaraldehyde in sodium cacodylate buffer (0.1 M, pH 7.2) at 4 °C. The dialysis membrane was then removed, and the cover slip was gently pulled out from the ring. To dehydrate the cells, the cover slip underwent a series of washes in increasing concentrations of ethanol (25, 50, 75, and 96%) for 10 min each time, then again in the same concentrations of Freon diluted in ethanol. Samples were observed using Through-Lens Detector operated at Secondary Electron (TLD-SE) mode by Magellan XHR SEM (FEI).

**Immuno-HR-SEM**. Bs cells were grown on EM grids (mesh copper grids, EMS) and were gently removed and grid-attached cells were washed three times with PBS×1, fixed with 2% paraformaldehyde and 0.01% glutaraldehyde in sodium cacodylate buffer (0.1 M, pH 7.2) for 10 min at 25 °C. Subsequently, grids were washed 3 times in PBS×1 and were incubated in PBS×1 containing 2% BSA and 0.1% Tween 20 for 30 min at 25 °C and washed twice with PBS×1. Next, grids were incubated for 2 h at 25 °C with rabbit anti-HA antibodies (Thermo Scientific, USA), diluted 1:1000 in PBSx1 containing 1% BSA. Grids were then washed three times with PBS×1 and incubated for 1 h at 25 °C with gold-conjugated goat anti-rabbit antibodies (Jackson, USA), diluted 1:500 in PBS×1. Grids were washed three times with PBS×1 and fixed with 2.5% glutaraldehyde in sodium cacodylate buffer (0.1 M, pH 7.2) for 1 h at 25 °C. Grids were then washed gently with water, and cells were dehydrated by exposure to a graded series of ethanol washes (25, 50, 75, 95, and 100% (×2); 10 min each), followed by overnight incubation in vacuum. Specimens were imaged without coating by Magellan XHR SEM (FEI) using Through-Lens Detector operated at Secondary Electron (TLD-SE) and Low-voltage high-Contrast backscatter electron Detector (vCD).

**RNA isolation and qRT-PCR**. RNA was extracted from Bs cells grown to the mid logarithmic phase by FastRNA Pro Blue kit (MP Biomedicals) according to the manufacturer protocol. RNA concentration was determined using NanoDrop 2000C (Thermo Scientific). RNA (2 µg) from each sample was treated with RQ1 DNase (2 Units, Promega), and subjected to cDNA synthesis using qScript cDNA synthesis kit (Quanta Biosciences), according to the manufacturer protocol. qRT-PCR reactions were conducted using iTaq Universal SYBR Green Supermix (Bio-Rad), and fluorescence detection was performed using Bio-Rad CFX Connect Real time system according to manufacturer instructions. RNA from rpsE and 16S rRNA was used to normalize expression. To verify that a single product was amplified a melt curve analysis was done using the Bio-Rad CFX manager software (v. 3.1). The relative gene expression levels were calculated from threshold cycle ($C_T$) values using the $2^{-\Delta\Delta CT}$ method[39]. Each assay was performed in duplicates with at least two RNA templates prepared from independent biological repeats. qRT-PCR primers were designed using Primer3 software (v. 0.4.0, available online).

**Data availability**. The authors declare that the relevant data supporting the findings of the study are available in this article and its Supplementary Information files, or from the corresponding author upon request.

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

## Acknowledgements

We thank I. Popov and E. Blayvas (NanoCentre, Hebrew U) for help with the EM analysis and A. Rouvinsky, I. Rosenshine and G. Bachrach (Hebrew University, IL), and members of the Ben-Yehuda laboratory for valuable comments. We are grateful to R. Biedendieck (Technische Universität Braunschweig, DE) for providing *Bm* plasmids. This work was supported by the European Research Council Advance Grant (339984) awarded to S.B.-Y.

## Author contributions

O.S., A.K.B. and S.B. performed the experiments. O.S., A.K.B., S.B. and S.B.-Y. conceived the experiments, analyzed the data, and wrote the manuscript. S.B.-Y. managed the project. G.B.M.M., E.T., L.S. and G.M. contributed tools, strains and reagents.

## Additional information

**Competing interests:** The authors declare no competing financial interests.

