## [Peer Review file · Nature Communications]

Reviewers' comments:

Reviewer #1 (Remarks to the Author):

Stempler and coworkers describe in this manuscript the fascinating role of nanotubes in the delivery of the toxin WapA. In addition, they employ a novel protein labelling method to show that nanotubes enable the transport of amino acids between cells.

The paper is clearly written and the importance of the protein YmdB, essential for nanotube formation, for the killing of *Bacillus megaterium* by *B. subtilis* is apparent. The fact that cell-cell contact is essential for this killing is also clearly shown. However, I am not convinced that their AHA-labelling experiment provides sufficient evidence for amino acid extraction from cells by nanotubes. I tried hard, but I could not follow their reasoning. In Fig 7B they show that a *Bs-metE-ymdB* mutant becomes green because it takes up AHA from the medium since it cannot take up methionine from the *Bm* strain, and cannot grow (like in Fig. 4C). However, in Fig. 5A a *Bs-metE* strain also becomes green and in this case nanotubes are being formed and the cells should be able to extract methionine from *Bm* cells since they grow. How can this be explained? In relation to this, I also do not understand why a *Bs-metE* strain does not take up AHA when mixed with *Bm* strain grown in the absence of methionine, whereas this is not the case when the *Bm* strain is pre-grown in medium with methionine. In both cases the *Bs-metE* strain can grow, thus using methionine from the *Bm* strains. Presumably, I am missing the point, but then the authors should thoroughly rephrase this part of the paper, including why the AHA-labelling method visualizes amino acid exchange, as this is also not clear to me.

Minor comments:

Figure 5D. It is unclear what these values mean since a ratio is given, yet the unit is given in CFU/ml.

Figure 7C-1. Why is the yellow cell a *B. megaterium* cell? It seems to have the same size as the *B. subtilis* cells.

Reviewer #2 (Remarks to the Author):

This manuscript by Stempler et al. investigates the interaction between *Bacillus subtilis* (*Bs*) and *Bacillus megaterium* (*Bm*). The authors show that *Bs* kills *Bm* in a contact-dependent manner, and they show that *wapA* and *ymdB* are required for killing of *Bm* by *Bs*. The authors also use the methionine analog L-azidohomoalanine (AHA) to show that *Bs* can obtain nutrients (methionine, specifically) from the *Bm* cells with which it interacts, and that this uptake of methionine requires cell-cell contact and also *ymdB*, which they had shown previously is required for nanotube formation. These are the main findings of the study.

The authors showed previously that *Bs* forms nanotubes between adjacent cells in a *ymdB*-dependent manner and that molecules (such as GFP) could be transferred between cells via the nanotubes. They also showed previously that *Bs* could transfer GFP to *Staphylococcus aureus*. Therefore, although this manuscript may be the first to describe and visualize interspecies "nutrient extraction", this finding is not particularly novel as the authors have already shown interspecies transfer of GFP (transfer of an amino acid is not that different).

The second main conclusion drawn by the authors is that WapA is delivered from one cell to another via the nanotubes. Although the data are consistent with this conclusion, they do not prove it and, in fact, equally plausible alternate explanations exist. Moreover, the authors did not, as stated in their title, visualize toxin delivery between bacteria – their data simply show that *Bs*

kills Bm in a wapA- and ymdB-dependent manner.

The authors' conclusion that WapA is delivered from Bs to Bm via the nanotube is based on the fact that both wapA and ymdB are required for Bs to kill Bm, and that ymdB is required for nanotube formation. However, other explanations for the data exist. For example, as YmdB is a phosphodiesterase involved in regulating the expression of many genes, it is possible that WapA delivery is independent of nanotube formation, but that wapA expression requires ymdB and hence a ymdB mutant cannot kill Bm simply because wapA is not expressed. Another possibility is that killing of Bm by Bs requires WapA and another molecule that is dependent on ymdB. To determine if WapA is sufficient for killing of Bm, the authors could express it, with and without the cognate immunity-encoding gene wapI, in Bm. If it is sufficient, expression of wapA alone will kill the bacteria while expression of wapA plus wapI will not. Determining if WapA is delivered through the nanotube will be more difficult, but if WapA were found to be transferred through nanotubes, it would raise the significance of the results substantially. Without such data, the advance in understanding from this work is modest.

Specific comments:

Page 1, line 4: The title is not consistent with the data presented

Page 2, lines 7-8: the authors did not show delivery of WapA from Bs to Bm

Page 3, line 17: bacterium should be bacterium's

Page 4, line 6: The authors did not show delivery of WapA to Bm. (Showing that Bs does not kill Bm that expresses wapI would support the authors statement.)

Page 6, line 11: The authors did not show delivery of WapA from Bs to Bm.

Page 7, line 21: The authors have not revealed that Bs is capable of delivering the WapA toxin into Bm.

Page 10, line 11: The authors have not shown that YmdB is required for delivery of toxin to neighboring cells.

Page 13, lines 1-2: The authors have shown that wapA is require for killing of Bm by Bs, but have not shown that "delivery of the WapA toxin is responsible for the observed Bm growth inhibition by Bs." And they have certainly not shown that WapA is delivered through the nanotube. These conclusions are overstated.

Reviewer #3 (Remarks to the Author):

Visualizing interspecies nutrient extraction and toxin delivery between bacteria

In this paper the authors investigate two phenomena in Bacillus; contact-dependent inhibition of cell growth and metabolic cross-feeding through nanotubes. Not surprisingly, the authors find that the contact-dependent growth inhibition between the two Bacillus species is mediated by WapA, a protein previously shown to function in contact-dependent growth inhibition in Bacillus. To expand the story, the authors look for additional factors required for the inhibition by screening 27 existing mutants, and find that the inhibition requires a YmdB. This leads the story to the investigation of cross-feeding through nanotubes, as the phosphodiesterase YmdB has previously been shown to be required for nanotube formation. The authors set up a novel and nifty approach to study metabolic cross-feeding, allowing detection of cross-feeding through fluorescence microscopy. In

general the paper is well written and in the scope of the journal, but I have some major concerns about what conclusions can be drawn from the study.

Major concerns

1. It is interesting that YmdB is required for both cross-feeding and contact-dependent growth inhibition, where the latter in particular has been studied to very small extent. My major concern is that the paper does not address why YmdB is required for these functions. YmdB is a global regulator and as such the YmdB effect on growth inhibition (and cross-feeding) could be through down-regulation of WapA expression. YmdB has been shown to regulate many genes in the Spo0A and SigD regulons, which both control WapA levels in the cell (Garti-Levi et al. J. bacteriology 2013 and for example Microbial Proteomics: Functional Biology of Whole Organisms or Antelmann et al. Proteomics 2002). I think it is most essential to find out if the YmdB mutation affects WapA levels either by transcriptional regulation (qPCR would answer this) or protein level (SigD controlled wall-associated proteases have previously been shown to degrade WapA. A western blot would show if this is the case.). In addition, over-expression of WapA in a strain lacking YmdB would tell if YmdB is actually required for WapA delivery.

2. Although it is interesting that YmbD controls both nutrient scavenging through nanotubes as well as contact-dependent growth inhibition, it also raises some concerns that need to be discussed in the paper. Mainly, why would it be a good idea to stop the growth of the cells that you want to get nutrients from? Wouldn't this limit the population of potential food source in the environment? Or is it the cell that is delivering nutrients whose growth is inhibited? This also doesn't make sense. Why would you stop the growth of the cell that is providing you with resources? Unless you are keeping it at a viable enough state so that it can produce the resource you want but not divide. A discussion of the relevance of the findings should be added.

3. The authors seem to suggest that nanotubes are required for WapA delivery. This is one of the most interesting aspects of the story, but unfortunately there is absolutely no data regarding this in the actual manuscript. Previous experiments from the lab have shown YmdB to be found in nanotubes. A similar experiment with WapA would show if WapA is indeed delivered through the tubes and would strengthen the paper substantially.

4. Finally, the contact-dependence assays lack proper controls. A larger filter allowing cells to pass freely between the compartments should be used to rule out the possibility that secreted molecules are bound to the membrane.

Minor points

1. Fig 2 is quite difficult to understand. Would it be possible to show the illustrations of the wells below the bars in the bar charts to make it easier to understand what is shown in each bar?

2. Experiment described on page 9, lanes 6-9 and in Fig 5C. When were the cells labeled with AHA? This is not clear either from the text, figure legend or materials and methods. Please specify.

3. Introduction page 4, line 8. "We further present evidence that these predatory activities are contact dependent and mediated by the phosphodiesterase YmdB," This should be rephrased to: "We further present evidence that these predatory activities are contact dependent and require the phosphodiesterase YmdB,". There is no data that show that YmdB actually mediates either nanotube formation or predation, it is required yes, but that could be due to regulation of expression of other genes.

4. Discussion page 13 lines 5-7. "WapA was shown to harbor a secretion signal sequence that

could potentially serve as a recognition signal for delivery. This characteristics hints that there is specificity in WapA molecular delivery, and that nanotubes could provide the route for its release. " This section should be removed. WapA contains a secretion signal for general secretion and has been shown to be linked to the cell-wall of *Bacillus subtilis*. The nanotubes have been shown to transfer cytoplasmic content between cells and how WapA, which is normally found in the cell-wall, would end up in the nanotubes is not obvious. At this point, there is no evidence that WapA is delivered through nanotubes.

Response to the Referees

We thank the Referees for the helpful and constructive comments regarding our manuscript: “Visualizing Interspecies Nutrient Extraction and Toxin Delivery between Bacteria” (Manuscript NCOMMS-16-27935), which we submitted for publication in *Nature Communications*. In the revised manuscript, we attempted to address the concerns raised by the Referees. Below we provide a summary of our major achievements and modifications, followed by a detailed point-by-point response to the raised comments.

Summary of the major changes:

1) A major concern raised by the referees was that there is no clear evidence that WapA is sufficient for killing *Bm*. To address this concern, we cloned the *Bs wapI* anti-toxin in *Bm*. The anti-toxin provided full immunity to *Bm* in mixed cultures as indicated by CFU assays and time lapse microscopy. These data are now presented in Figure S3B-S3C and described in the text (p6 lines 23-25). Of note, we have cloned the *wapI* gene in an additional *Bm* strain and obtained similar results, reinforcing the view that in general *Bs* kills *Bm* by delivering WapA. We thank the Reviewers for this helpful suggestion.

2) The Reviewers were concerned that the effect of YmdB on WapA delivery could be indirect, as YmdB was previously shown to affect the transcription of many genes. This is indeed a valid concern. To address this issue, we carried out qRT-PCR, as suggested by the Referees, and could not detect any significant difference in *wapA* transcription in the presence or the absence of *ymdB*. Furthermore, we constructed a P_{wapA} -*gfp* reporter to visualize *wapA* transcription in individual cells, and again could not detect any significant difference. Taken together, YmdB does not affect either the transcription of *wapA* or its steady state levels. These data are now included in Figure 3C and Figure S4A, and are discussed in the text (p7 lines 16-21).

3) The Reviewers pointed out that we did not show directly that the WapA toxin is delivered by nanotubes. To address this concern we constructed a *Bs* strain harboring WapA tagged with the HA epitope and carried out immuno-HR-SEM. Remarkably, we could visualize WapA molecules that were localized directly to nanotubes. These results are now included in Figure 3D and Figure S4B, and are discussed in the text (p7 line 21- p8 line 9). These results directly show that nanotubes provide a route for WapA delivery.

Of note, we have previously identified WapA in a group of 50 proteins that were enriched in the nanotube biochemical fraction (Dubey et al., 2016; Table S1). We have now mentioned this previous finding in the text (p7 lines 24-25).

Reference

Dubey, G.P., Malli Mohan, G.B., Dubrovsky, A., Amen, T., Tsipshtein, S., Rouvinski, A., Rosenberg, A., Kaganovich, D., Sherman, E., Medalia, O., *et al.* (2016). Architecture and Characteristics of Bacterial Nanotubes. *Developmental cell* 36, 453-461.

Point-by-point response to the Reviewers

Reviewer #1 (Remarks to the Author):

Stempler and coworkers describe in this manuscript the fascinating role of nanotubes in the delivery of the toxin WapA. In addition, they employ a novel protein labelling method to show that nanotubes enable the transport of amino acids between cells. The paper is clearly written and the importance of the protein YmdB, essential for nanotube formation, for the killing of *Bacillus megaterium* by *B. subtilis* is apparent. The fact that cell-cell contact is essential for this killing is also clearly shown.

1) However, I am not convinced that their AHA-labelling experiment provides sufficient evidence for amino acid extraction from cells by nanotubes. I tried hard, but I could not follow their reasoning. In Fig 7B they show that a *Bs-metE-ymdB* mutant becomes green because it takes up AHA from the medium since it cannot take up methionine from the *Bm* strain, and cannot grow (like in Fig. 4C). However, in Fig. 5A a *Bs-metE* strain also becomes green and in this case nanotubes are being formed and the cells should be able to extract methionine from *Bm* cells since they grow. How can this be explained?

Unfortunately, the Reviewer missed a very critical point in our experimental design of the amino acid extraction experiments, which is the pre-growth of the cells with or without Met prior to strain mixing.

The difference between Fig 5A and 7B is the pre-growth of the *Bm* cells. In Fig 5A, *Bm* was pre-grown in Met, hence the cells were prone to take up Met from the medium followed by AHA uptake, according to the assay (see Fig 4 for details). The correct comparison should be between Fig 5B and Fig 7B. In the experiments described in both figures *Bm* cells were pre-grown without Met, thus making their own Met, and the cells therefore remained unlabelled by the less preferred AHA (see Fig 4 for details).

To avoid such confusion by the readers we have now indicated in the text that the results in Fig 7A are similar to those in Fig 5B (p10 lines 20-21).

2) In relation to this, I also do not understand why a *Bs-metE* strain does not take up AHA when mixed with *Bm* strain grown in the absence of methionine, whereas this is not the case when the *Bm* strain is pre-grown in medium with methionine. In both cases the *Bs-metE* strain can grow, thus using methionine from the *Bm* strains. Presumably, I am missing the point, but then the authors should thoroughly rephrase this part of the paper, including why the AHA-labelling method visualizes amino acid exchange, as this is also not clear to me.

Again, this is an important point, described in detail in Fig 4. In the presence of external Met the cells express Met transporters, and therefore the replacement of external Met by AHA leads to strong labelling. This is explained in the text in p8 line 10- p9 line 7.

Minor comments:

Figure 5D. It is unclear what these values mean since a ratio is given, yet the unit is given in CFU/ml.

Thank you, CFU/ml was removed.

Figure 7C-1. Why is the yellow cell a *B. megaterium* cell? It seems to have the same size as the *B. subtilis* cells.

We have carefully inspected many similar images and we could separate *Bm* from *Bs* with relative ease. In general, *Bm* cells look larger and thicker than *Bs*, and due to the size difference *Bm* cells also appear brighter in non-coated SEM samples. In this particular image, the yellow cell looks larger, thicker and brighter than the rest of the cells. It might be confusing because *Bs* cells in this picture are in chains whereas *Bm* is only a single cell. Some differences in size could also be attributed to fixation, required for SEM. We have carefully analysed our images, and we are convinced that our interpretation is correct. We also present here the original SEM micrograph of this specific image, before false colouring, to strengthen our point.

Original SEM micrograph for Figure 7C1.

Reviewer #2 (Remarks to the Author):

This manuscript by Stempler et al. investigates the interaction between *Bacillus subtilis* (*Bs*) and *Bacillus megaterium* (*Bm*). The authors show that *Bs* kills *Bm* in a contact-dependent manner, and they show that *wapA* and *ymdB* are required for killing of *Bm* by *Bs*. The authors also use the methionine analog L-azidohomoalanine (AHA) to show that *Bs* can obtain nutrients (methionine, specifically) from the *Bm* cells with which it interacts, and that this uptake of methionine requires cell-cell contact and also *ymdB*, which they had shown previously is required for nanotube formation. These are the main findings of the study.

1) The authors showed previously that *Bs* forms nanotubes between adjacent cells in a *ymdB*-dependent manner and that molecules (such as GFP) could be transferred between cells via the nanotubes. They also showed previously that *Bs* could transfer GFP to *Staphylococcus aureus*. Therefore, although this manuscript may be the first to describe and visualize interspecies “nutrient extraction”, this finding is not particularly novel as the authors have already shown interspecies transfer of GFP (transfer of an amino acid is not that different).

We would like to highlight that as far as we know, even though some interspecies nutrient exchange has been reported, this is the first time that such a phenomenon has ever been visualized. Furthermore, nutrient extraction acts in opposing directionality; so far we have shown the transfer of molecules from donor to recipient, while here we show the opposite action in which one bacterium preys on and extracts nutrient from an opponent strain. This assay also reflects a naturally occurring process, while GFP delivery is artificial.

2) The second main conclusion drawn by the authors is that WapA is delivered from one cell to another via the nanotubes. Although the data are consistent with this conclusion, they do not prove it and, in fact, equally plausible alternate explanations exist. Moreover, the authors did not, as stated in their title, visualize toxin delivery between bacteria – their data simply show that Bs kills Bm in a *wapA*- and *ymdB*-dependent manner.

We now provide evidence for the direct localization of WapA to nanotubes as described in major changes, point 3.

3) The authors' conclusion that WapA is delivered from Bs to Bm via the nanotube is based on the fact that both *wapA* and *ymdB* are required for Bs to kill Bm, and that *ymdB* is required for nanotube formation. However, other explanations for the data exist. For example, as *YmdB* is a phosphodiesterase involved in regulating the expression of many genes, it is possible that WapA delivery is independent of nanotube formation, but that *wapA* expression requires *ymdB* and hence a *ymdB* mutant cannot kill Bm simply because *wapA* is not expressed.

To address this possibility, we carried out qRT-PCR and used a P_{wapA} -*gfp* reporter to assay *wapA* transcription. *YmdB* did not affect the transcription of *wapA*. These data are now included in the manuscript (see major changes, point 2).

4) Another possibility is that killing of Bm by Bs requires WapA and another molecule that is dependent on *ymdB*. To determine if WapA is sufficient for killing of Bm, the authors could express it, with and without the cognate immunity-encoding gene *wapI*, in Bm. If it is sufficient, expression of *wapA* alone will kill the bacteria while expression of *wapA* plus *wapI* will not.

To address this concern, we have now cloned the *wapI* anti-toxin from *Bs* in *Bm*, and could see that it provides full immunity. These results are now included in the manuscript (see major changes, point 1).

5) Determining if WapA is delivered through the nanotube will be more difficult, but if WapA were found to be transferred through nanotubes, it would raise the significance of the results substantially. Without such data, the advance in understanding from this work is modest.

Using immuno-HR-SEM with antibodies against HA tagged WapA, we now provide evidence for the direct localization of WapA to nanotubes (see major changes, point 3).

Specific comments:

Page 1, line 4: The title is not consistent with the data presented

We now visualized the direct localization of WapA to nanotubes (see major changes, point 3). Therefore, we think that the title now better reflects our findings.

Page 2, lines 7-8: the authors did not show delivery of WapA from Bs to Bm

According to the Reviewer's suggestion, we now provide substantiating evidence that WapA is responsible for *Bm* death (see major changes, point 1). We therefore think it is reasonable to claim that "*Bs* was found to rapidly inhibit *Bm* growth by delivering the tRNase toxin WapA".

Page 3, line 17: bacterium should be bacterium's

Thank you, corrected.

Page 4, line 6: The authors did not show delivery of WapA to Bm. (Showing that Bs does not kill Bm that expresses wapI would support the authors statement.)

According to the Reviewer's suggestion, we now provide substantiating evidence that WapA is responsible for *Bm* death (see major changes, point 1).

Page 6, line 11: The authors did not show delivery of WapA from Bs to Bm.

According to the Reviewer's suggestion, we now provide substantiating evidence that WapA is responsible for *Bm* death (see major changes, point 1).

Page 7, line 21: The authors have not revealed that Bs is capable of delivering the WapA toxin into Bm.

According to the Reviewer's suggestion, we now provide substantiating evidence that WapA is responsible for *Bm* death (see major changes, point 1).

Page 10, line 11: The authors have not shown that YmdB is required for delivery of toxin to neighboring cells.

The claim that: "The observation that YmdB is required for exchange of toxin and nutrients between the two *Bacilli*", was toned down to: "The observation that the toxin delivery and nutrient extraction are dependent on YmdB" (p11, line 3).

Page 13, lines 1-2: The authors have shown that wapA is require for killing of Bm by Bs, but have not shown that "delivery of the WapA toxin is responsible for the observed Bm growth inhibition by Bs." And they have certainly not shown that WapA is delivered through the nanotube. These conclusions are overstated.

According to the Reviewer's suggestion, we now provide substantiating evidence that WapA is responsible for *Bm* death (see major changes, point 1), and that WapA localizes to nanotubes (see major changes, point 3).

Reviewer #3 (Remarks to the Author):

Visualizing interspecies nutrient extraction and toxin delivery between bacteria

In this paper the authors investigate two phenomena in *Bacillus*; contact-dependent inhibition of cell growth and metabolic cross-feeding through nanotubes. Not surprisingly, the authors find that the contact-dependent growth inhibition between the two *Bacillus* species is mediated by WapA, a protein previously shown to function in contact-dependent growth inhibition in *Bacillus*. To expand the story, the authors look for additional factors required for the inhibition by screening 27 existing mutants, and find that the inhibition requires a YmdB. This leads the story to the investigation of cross-feeding through nanotubes, as the phosphodiesterase YmdB has previously been shown to be required for nanotube formation. The authors set up a novel and nifty approach to study metabolic cross-feeding, allowing detection of cross-feeding through fluorescence microscopy. In general the paper is well written and in the scope of the journal, but I have some major concerns about what conclusions can be drawn from the study.

Major concerns

1. It is interesting that YmdB is required for both cross-feeding and contact-dependent growth inhibition, where the latter in particular has been studied to very small extent. My major concern is that the paper does not address why YmdB is required for these functions. YmdB is a global regulator and as such the YmdB effect on growth inhibition (and cross-feeding) could be through down-regulation of WapA expression. YmdB has been shown to regulate many genes in the Spo0A and SigD regulons, which both control WapA levels in the cell (Garti-Levi et al. J. bacteriology 2013 and for example Microbial Proteomics: Functional Biology of Whole Organisms or Antelmann et al. Proteomics 2002). I think it is most essential to find out if the YmdB mutation affects WapA levels either by transcriptional regulation (qPCR would answer this) or protein level (SigD controlled wall-associated proteases have previously been shown to degrade WapA. A western blot would show if this is the case.). In addition, over-expression of WapA in a strain lacking YmdB would tell if YmdB is actually required for WapA delivery.

To address this possibility, we carried out qRT-PCR and used a P_{wapA} -*gfp* reporter to assay *wapA* transcription. YmdB did not affect the transcription of *wapA* (thus, there was no need to overexpress *wapA* in the mutant cells). These data are now included in the manuscript (see major changes, point 2).

2. Although it is interesting that YmbD controls both nutrient scavenging through nanotubes as well as contact-dependent growth inhibition, it also raises some concerns that need to be discussed in the paper. Mainly, why would it be a good idea to stop the growth of the cells that you want to get nutrients from? Wouldn't this limit the population of potential food source in the environment? Or is it the cell that is delivering nutrients whose growth is inhibited? This also doesn't make sense. Why would you stop the growth of the cell that is providing you with resources? Unless you are keeping it at a viable enough state so that it can

produce the resource you want but not divide. A discussion of the relevance of the findings should be added.

This is an interesting point that could be the explanation for interspecies cross feeding interactions (e.g; Benomar et al., 2015). However, we attempted to provide a model that explains our current findings. In our view, the most important activity for the bacterium is to inhibit the growth of its opponent that occupies the same niche and competes for the same resources. After inhibiting the growth of its niche rival, the bacterium can further benefit from exploiting its opponent's vital resources. This capability can facilitate growth and division of the predator temporally. This idea is now clarified in the discussion (p12 lines 22-23).

3. The authors seem to suggest that nanotubes are required for WapA delivery. This is one of the most interesting aspects of the story, but unfortunately there is absolutely no data regarding this in the actual manuscript. Previous experiments from the lab have shown YmdB to be found in nanotubes. A similar experiment with WapA would show if WapA is indeed delivered through the tubes and would strengthen the paper substantially.

Using immuno-HR-SEM with antibodies against HA tagged WapA, we now provide evidence for the direct localization of WapA to nanotubes (see major changes, point 3).

4. Finally, the contact-dependence assays lack proper controls. A larger filter allowing cells to pass freely between the compartments should be used to rule out the possibility that secreted molecules are bound to the membrane.

In our system, using larger filter pores would have been insufficient to allow the free diffusion of the cells as *Bm* is relatively a large bacterium (with a width of 0.8-1 μm), and both *Bs* and *Bm* forms elongated chains (typically over 20 μm) (see for example Fig 1B). To address the Reviewer's concern that the membrane interferes with the toxic activity of *Bs* on *Bm*, we grew both strains in the same compartment and the toxic effect was restored. These results are now shown in Fig S3A, and included in the text (p6 lines 8-9).

Minor points

1. Fig 2 is quite difficult to understand. Would it be possible to show the illustrations of the wells below the bars in the bar charts to make it easier to understand what is shown in each bar?

We considered carefully the Reviewer's suggestion; however, since the comparison is between bacteria located in different chambers (not only different compartments), we think that the way we originally presented the data is less confusing. We made minor changes in Fig 2B to make it clearer.

2. Experiment described on page 9, lanes 6-9 and in Fig 5C. When were the cells labeled with AHA? This is not clear either from the text, figure legend or materials and methods. Please specify.

We added to the main text that the cells were incubated "in medium containing AHA" (p9, line 20)". This information is also specified in the legend "Both the mixture in the inner well, and the *Bs* grown alone in the outer well, shared the same S7 medium containing AHA (1 mM)". Finally, we further clarified the obtained results in the text (p9, line 25- p10, line 1).

3. Introduction page 4, line 8. "We further present evidence that these predatory activities are contact dependent and mediated by the phosphodiesterase YmdB," This should be rephrased to: "We further present evidence that these predatory activities are contact dependent and require the phosphodiesterase YmdB,". There is no data that show that YmdB actually mediates either nanotube formation or predation, it is required yes, but that could be due to regulation of expression of other genes.

Modified.

4. Discussion page 13 lines 5-7. "WapA was shown to harbor a secretion signal sequence that could potentially serve as a recognition signal for delivery. This characteristics hints that there is specificity in WapA molecular delivery, and that nanotubes could provide the route for its release. "This section should be removed. WapA contains a secretion signal for general secretion and has been shown to be linked to the cell-wall of *Bacillus subtilis*. The nanotubes have been shown to transfer cytoplasmic content between cells and how WapA, which is normally found in the cell-wall, would end up in the nanotubes is not obvious. At this point, there is no evidence that WapA is delivered through nanotubes.

We agree that this was a relatively speculative part, and accordingly the section on WapA in the discussion was revised to accommodate the relevant information and our new data, cite the corresponding papers, and rephrase our speculations (p13 lines 6-10).

References

Benomar, S. *et al.* Nutritional stress induces exchange of cell material and energetic coupling between bacterial species. *Nature communications* 6, 6283, doi:10.1038/ncomms7283 (2015).

Dubey, G. P. *et al.* Architecture and Characteristics of Bacterial Nanotubes. *Developmental cell* 36, 453-461, doi:10.1016/j.devcel.2016.01.013 (2016).

REVIEWERS' COMMENTS:

Reviewer #1 (Remarks to the Author):

After reading the revision of the paper by Stempler and coworkers I am stuck with the same question as I had the first time, which is their peculiar behaviour of their metE deletion strain on which important conclusions are based.

I will try to explain why I cannot follow their reasoning (after trying hard). The issue is related to Fig 4 and Fig. 5.

In Fig. 4 the experimental setup for AHA labelling is explained. The idea is that when the external concentration of Met (methionine) is high, the Met synthetic gene cluster is switched off and the Met-uptake system is switched on and cells are labelled by AHA. In Fig. 4C the authors introduce the metE deletion strain which is unable to synthesise methionine. Therefore, this strain produces higher levels of Met-transporter than wild type cells grown in the presence of Met. Logically, because they cannot make Met themselves and have to take Met from the medium. Therefore, the metE mutant shows a higher AHA signal than wild type cells pregrown in Met-containing medium (Fig. 4A & C, Fig. S5).

In Fig. 5A the metE mutant (pregrown in Met-containing medium) is mixed with Bm pregrown in Met medium and incubated in the presence of AHA. Both strains show a clear AHA signal. This is not surprising because like Fig 4C, the metE mutant has expressed Met-transporters and will readily take up AHA, and so does the Bm strain pregrown in Met medium (like in Fig. 4A).

The peculiar and intriguing thing happens when the metE mutant (pregrown in Met-containing medium) is mixed with a Bm strain that was pregrown in medium without Met (Fig. 5B). Then both strains do NOT show AHA incorporation. I do understand why the Bm strain is not labelled, since it has not expressed Met-transporters (due to pregrowth in medium lacking Met, as is shown in Fig. 4B). However, I do not understand why the metE mutant is not labelled by AHA, because this strain has expressed Met-transporters during pregrowth in Met-containing medium (Fig. 4C). So how is it possible that these transporters are suddenly no longer functional when mixed with these Bm cells? Why would the metE mutant suddenly not be able to take up AHA from the medium, whereas in Fig. 4C this happened easily? Does this mean that the Met uptake system is switched off? But why and how? Something interesting is going on, but I do not see how the sudden absence of AHA uptake in the metE mutant can be translated to 'nutrient extraction'. The same puzzlement arises for the same reasons when studying Fig. 6 and Fig. 7.

Something is going on that seems to be contact (YmdB) dependent, but I cannot see how these results are proof for a transfer of nutrients between cells. It might be that by feeding Met from Bm cells to the metE mutant there is sufficient methionine for growth but such a low level that the Met transporter expression is reduced substantially. If that is the case then the authors should prove this (and for sure discuss this). Because Fig. 5A is not the correct control since this is basically the same condition as in Fig. 4A where both strains were pregrown in Met medium.

Unfortunately, the authors have not taken away my doubts about their explanation in their revision. But again, I may be mistaken and have missed the point.

Reviewer #2 (Remarks to the Author):

Stempler et al. have addressed the concerns of the referees adequately. The results of the additional experiments included in the revised manuscript justify most of the conclusions that were not completely justified before. Some others were softened such that the manuscript is, in my opinion, ready for publication. I have only a few minor comments (some VERY minor) for the authors to consider:

1. Page 3, line 9: insert "a" between "as" and "tubular"
 2. Page 3, line 16 and elsewhere: should be "contact-dependent" (i.e., with a hyphen)
 3. Page 4, lines 6-7: should read ". . . and extracts nutrients from, Bacillus megaterium (Bm), a neighboring target bacterium, exhibiting. . ."
 4. Page 6, line 25: this is up to the authors, but why not include the data from Fig. S3B-S3C in Fig. 1?
 5. Page 8, top of page: Doesn't the fact that WapA is detectable by antibodies in this experiment mean that it is on the outside of the nanotubes? This point should be discussed. Moreover, the experiment would be even more convincing if the interaction was between Bs and Bm – did the authors try to visualize this?
- Page 13: The authors should expand their discussion about whether WapA is being delivered through the nanotubes or somehow on the outside of the nanotubes. Perhaps a bit of speculation about how this might occur if it is via the outside?

Response to the Referees

We thank you and the Referees for the helpful and constructive comments regarding our manuscript. Below is a detailed point-by-point response to the comments raised by the referees.

Reviewer #1 (Remarks to the Author)

After reading the revision of the paper by Stempler and coworkers I am stuck with the same question as I had the first time, which is their peculiar behaviour of their metE deletion strain on which important conclusions are based.

I will try to explain why I cannot follow their reasoning (after trying hard). The issue is related to Fig 4 and Fig. 5.

In Fig. 4 the experimental setup for AHA labelling is explained. The idea is that when the external concentration of Met (methionine) is high, the Met synthetic gene cluster is switched off and the Met-uptake system is switched on and cells are labelled by AHA. In Fig. 4C the authors introduce the metE deletion strain which is unable to synthesise methionine. Therefore, this strain produces higher levels of Met-transporter than wild type cells grown in the presence of Met. Logically, because they cannot make Met themselves and have to take Met from the medium. Therefore, the metE mutant shows a higher AHA signal than wild type cells pregrown in Met-containing medium (Fig. 4A & C, Fig. S5).

In Fig. 5A the metE mutant (pregrown in Met-containing medium) is mixed with Bm pregrown in Met medium and incubated in the presence of AHA. Both strains show a clear AHA signal. This is not surprising because like Fig 4C, the metE mutant has expressed Met-transporters and will readily take up AHA, and so does the Bm strain pregrown in Met medium (like in Fig. 4A).

The peculiar and intriguing thing happens when the metE mutant (pregrown in Met-containing medium) is mixed with a Bm strain that was pregrown in medium without Met (Fig. 5B). Then both strains do NOT show AHA incorporation. I do understand why the Bm strain is not labelled, since it has not expressed Met-transporters (due to pregrowth in medium lacking Met, as is shown in Fig. 4B). However, I do not understand why the metE mutant is not labelled by AHA, because this strain has expressed Met-transporters during pregrowth in Met-containing medium (Fig. 4C). So how is it possible that these transporters are suddenly no longer functional when mixed with these Bm cells? Why would the metE mutant suddenly not be able to take up AHA from the medium, whereas in Fig. 4C this happened easily? Does this mean that the Met uptake system is switched off? But why and how? Something interesting is going on, but I do not see how the sudden absence of AHA uptake in the metE mutant can be translated to 'nutrient extraction'. The same puzzlement arises for the same reasons when studying Fig. 6 and Fig. 7.

Something is going on that seems to be contact (YmdB) dependent, but I cannot see how these results are proof for a transfer of nutrients between cells. It might be that by feeding Met from Bm cells to the metE mutant there is sufficient methionine for

growth but such a low level that the Met transporter expression is reduced substantially. If that is the case then the authors should prove this (and for sure discuss this). Because Fig. 5A is not the correct control since this is basically the same condition as in Fig. 4A where both strains were pregrown in Met medium.

Unfortunately, the authors have not taken away my doubts about their explanation in their revision. But again, I may be mistaken and have missed the point.

We thank the Reviewer for clarifying this issue. We think that the missing point by the Referee was that the cells were incubated with AHA for 1.5-2 hrs prior to the click reaction in all the described experiments (for single species and *Bs-Bm* mixed cultures). Therefore, the transporters for Met were not "suddenly no longer functional" as the Reviewer pointed out.

This information is specified in all relevant figure legends, and is now clearly included in the main text and the Methods. We also added a sentence stating (p9) that "Conceivably; the expression of the Met transporters in the *Bs* ($\Delta metE$) was decreased during the co-incubation period, resulting in unlabeled cells". Consistent with this idea, *Bs* ($\Delta metE$) could grow and divide in a medium lacking Met, only when co-cultured with *Bm* (Fig. 5D).

We hope that these additions and clarifications address the Reviewer's concerns.

Reviewer #2 (Remarks to the Author):

Stempler et al. have addressed the concerns of the referees adequately. The results of the additional experiments included in the revised manuscript justify most of the conclusions that were not completely justified before. Some others were softened such that the manuscript is, in my opinion, ready for publication. I have only a few minor comments (some VERY minor) for the authors to consider:

1. Page 3, line 9: insert "a" between "as" and "tubular"

Added, thank you.

2. Page 3, line 16 and elsewhere: should be "contact-dependent" (i.e., with a hyphen).

Modified, thank you.

3. Page 4, lines 6-7: should read ". . . and extracts nutrients from, *Bacillus megaterium* (*Bm*), a neighboring target bacterium, exhibiting. . ."

This part was modified according to the Editor's suggestion.

4. Page 6, line 25: this is up to the authors, but why not include the data from Fig. S3B-S3C in Fig. 1?

Thank you, we indeed thought about it and decided that Fig. 1 is too crowded.

5. Page 8, top of page: Doesn't the fact that WapA is detectable by antibodies in this experiment mean that it is on the outside of the nanotubes? This point should be discussed.

As the immuno-SEM analysis utilized in this study to visualize WapA molecules over nanotubes (Fig. 3D) could perturb nanotube integrity, the antibody signal can be

obtained from the nanotube lumen or surface. This point has now been included in the Discussion.

Moreover, the experiment would be even more convincing if the interaction was between Bs and Bm - did the authors try to visualize this?

Yes, indeed we tried to visualize both species by immuno-SEM. Unfortunately, *Bm* cells were very sensitive to the immuno-SEM procedure and were washed away from the grids, making the analysis very challenging.

Page 13: The authors should expand their discussion about whether WapA is being delivered through the nanotubes or somehow on the outside of the nanotubes. Perhaps a bit of speculation about how this might occur if it is via the outside?

We added a sentence to the Discussion speculating that WapA interaction with the nanotube membrane enables WapA translocation.